# InMu-Net: Advancing Multi-modal Intent Detection via Information Bottleneck and Multi-sensory Processing

Zhihong Zhu
Peking University
Beijing, China
zhihongzhu@stu.pku.edu.cn

Xuxin Cheng
Peking University
Beijing, China
chengxx@stu.pku.edu.cn

Zhaorun Chen
The University of Chicago
Chicago, USA
zhaorun@uchicago.edu

Yuyan Chen
Fudan University
Shanghai, China
chenyuyan21@m.fudan.edu.cn

Yunyan Zhang
Jarvis Research Center, Tencent
YouTu Lab
Shenzhen, China
yunyanzhang@tencent.com

Xian Wu*
Jarvis Research Center, Tencent
YouTu Lab
Shenzhen, China
kevinxwu@tencent.com

Yefeng Zheng
Medical Artificial Intelligence Lab,
Westlake University
Hangzhou, China
Jarvis Research Center, Tencent
YouTu Lab
Shenzhen, China
zhengyefeng@westlake.edu.cn

Bowen Xing
Beijing Key Laboratory of Knowledge
Engineering for Materials Science,
School of Computer and
Communication Engineering,
University of Science and Technology
Beijing
Beijing, China
bwxing714@gmail.com

## Abstract

Multi-modal intent detection (MID) aims to comprehend users' intentions through diverse modalities, which has received widespread attention in dialogue systems. Despite the promising advancements in complex fusion mechanisms or architecture designs, challenges remain due to: (1) various noise and redundancy in both visual and audio modalities and (2) long-tailed distributions of intent categories. In this paper, to tackle the above two issues, we propose InMu-Net, a simple yet effective framework for MID from the **In**formation bottleneck and **Mu**lti-sensory processing perspective. Our contributions lie in three aspects. First, we devise a *denoising bottleneck module* to filter out the intent-irrelevant information in the fused feature; Second, we introduce a *saliency preservation loss* to prevent the dropping of intent-relevant information; Ultimately, *kurtosis regulation* is introduced to maintain representation smoothness during the filtering process, mitigating the adverse impact of the long tail distribution. Comprehensive experiments on two MID benchmark datasets demonstrate the effectiveness of InMu-Net and its vital components. Impressively, a series of analyses reveal our denoising potential and robustness in low-resource, modality corruption, cross-architecture and cross-task scenarios.

## CCS Concepts

• **Computing methodologies** → **Discourse, dialogue and pragmatics**.

## Keywords

Multi-modal Intent Detection, Multi-modal Information Bottleneck, Multi-sensory Processing

**ACM Reference Format:**
Zhihong Zhu, Xuxin Cheng, Zhaorun Chen, Yuyan Chen, Yunyan Zhang, Xian Wu, Yefeng Zheng, and Bowen Xing. 2024. InMu-Net: Advancing Multi-modal Intent Detection via Information Bottleneck and Multi-sensory Processing. In *Proceedings of the 32nd ACM International Conference on Multimedia (MM '24), October 28-November 1, 2024, Melbourne, VIC, Australia.* ACM, New York, NY, USA, 10 pages. https://doi.org/10.1145/3664647.3681623

*Corresponding author.

## 1 Introduction

Intent detection (ID) aims to ascertain the objectives of users conveyed through their utterances, which serves as a crucial component of task-oriented dialogue systems [31, 37]. Prior studies have extensively researched ID and validated the significance of textual modality [26, 59]. However, beyond textual utterances, facial expressions and audio signals are also informative as they are often complementary and interact synergistically [50]. Therefore, multimodal intent detection (MID) has attracted increasing research attention, which is more practical in real-world scenarios.

To effectively leverage the information from various modalities, numerous methods have been proposed for MID. Therein, Saha

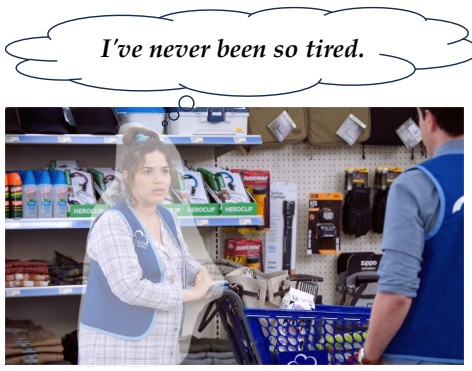

Figure 1: An example of redundancy in the visual modality in MIntRec [60] dataset. As indicated, the utterance is only relevant to the woman highlighted in gray, and the information like the distracting background is intent-irrelevant.

et al. [35], Zhang et al. [60] pioneered MID area by introducing tri-modal ID benchmarks with text, visual and audio information; Zhou et al. [61] proposed a token-level contrastive learning method with modality-aware prompting to facilitate modality fusion; Huang et al. [14] introduced a shallow-to-deep Transformer-based framework with ChatGPT-based data augmentation strategy to align different modality features, obtaining state-of-the-art (SOTA) results.

Despite promising advancements achieved, we discover that existing MID models still suffer from two main issues:

(1) **Noise and redundancy in visual and audio modalities.** As depicted in Figure 1, there exists plenty of redundancy and noise in the visual modality, which can impede the performance of multi-modal fusion. To this end, several studies proposed methods like gating mechanisms [21, 54] to filter noise and redundancy. However, an underexplored aspect is the potential of these fusion gates to filter vital information within the filtered modalities inadvertently.

(2) **Long-tailed distributions of intent categories.** Existing MID benchmarks [35, 60] exhibit a pronounced long-tailed distribution of intent categories as shown in Figure 2, where a few classes, *i.e.*, head classes, contain a major number of samples while the remaining classes, *i.e.*, tail classes, have only a small number of samples. A straightforward remedy is to rebalance the training dataset through weighted sampling. However, this is a suboptimal strategy that may be detrimental to the accuracy of the head classes [13, 49].

In this paper, we propose a new framework termed INMU-NET to address the existing issues jointly, drawing inspirations from **In**formation bottleneck [9, 48, 52] and **Mu**lti-sensory processing [28, 43]. Through three vital components, INMU-NET advances towards its denoising and redundancy reduction capabilities, alongside bolstering robustness across multiple categories. Specifically, ❶ we design a *denoising bottleneck module* to effectively reduce intent-irrelevant feature redundancy. ❷ we present a *saliency preservation loss*, which provides explicit supervision to maximize the intent-relevant information in the fused feature. ❸ we perform *kurtosis regulation* on both unimodal and multi-modal representations. In this manner, INMU-NET can diminish sensitivity towards tail intents, thus mitigating the adverse effects of long-tailed distribution.

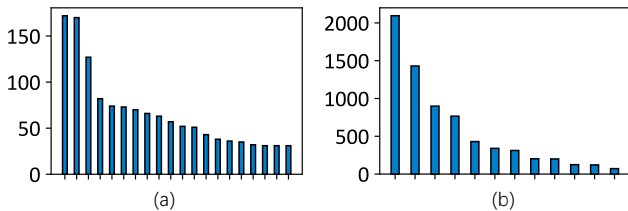

Figure 2: Distribution of intent categories on two MID training datasets: (a) MIntRec [60] and (b) MELD-DA [35].

Quantitative experiments demonstrate that our INMU-NET significantly outperforms previous SOTA methods. Systematic analyses confirm the superiority of INMU-NET against distinct scenarios.
**Contributions**. In a nutshell, our contributions are three-fold:
- We present a new framework dubbed INMU-NET for MID, drawing inspirations from the information bottleneck and multi-sensory processing. To our best knowledge, this is the first attempt to bridge the information bottleneck and MID.
- We introduce three core modules in the proposed INMU-NET from the principled perspective, addressing significant issues of noise redundancy and the long-tail problem in MID.
- Extensive experiments including low-resource, modality corruption, cross-architecture, and cross-task scenarios demonstrate the generalizability and robustness of our INMU-NET.

## 2 Preliminaries

### 2.1 Task Description

Formally, given a tri-modal input comprising text, visual and audio modalities, the multi-modal intent detection (MID) task can be conceptualized as a classification task that determines the intent label for the tri-modal input, which is expressed as follows:

$$y = f(X_t, X_v, X_a), \tag{1}$$

where $f(\cdot)$ represents the MID model; $X_t, X_v, X_a$ denote the text, visual and audio input, respectively; $y \in Y = \{y_1, y_2, \ldots, y_K\}$ is the intent label associated with one of the $K$ predefined intents.

### 2.2 Feature Encoding

To begin, we first encode the multi-modal sequential input $X_m$ (where $m \in \{t, v, a\}$) into unit-length representations $f_m$. Specifically, we employ separate modality-specific Transformers [45] to capture the features of distinct modalities as follows:

$$f_m = \text{Transformer}_m(X_m; \theta_m). \tag{2}$$

in which $\theta_m$ represents the parameters of the Transformer$_m$.

### 2.3 Mutual Information

Mutual information [17] is a measure of the amount of information shared between random variables. Formally, it quantifies the statistical dependency of two random variables $X$ and $Y$:

$$I(X; Y) = \mathbb{E}_{p(X,Y)} \left[ \log \frac{p(X, Y)}{p(X)p(Y)} \right], \tag{3}$$

where $p(X, Y)$ denotes the joint probability distribution between $X$ and $Y$, while $p(X)$ and $p(Y)$ are their marginals.

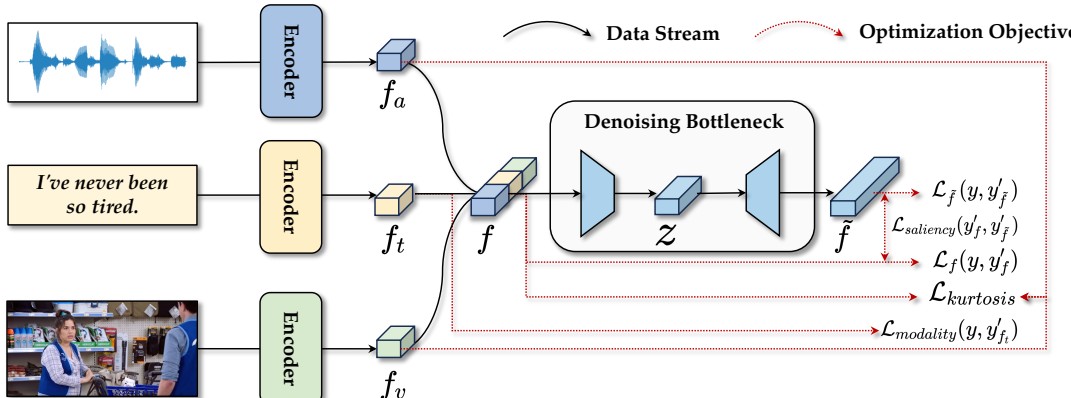

Figure 3: The architecture of InMu-Net. Mathematical symbols in the figure are consistent with the formulas in the paper.

## 2.4 Kurtosis Estimation

Kurtosis [6] is a statistical measure utilized to categorize the tail behavior of distributions. It is sensitive to rare events and is employed for analyzing distributions with "fatter tails". For *univariate* variables, kurtosis is quantified as the standardized fourth moment:

$$\frac{\mathbb{E}\left[(X-\mu)^4\right]}{(\mathbb{E}\left[(X-\mu)^2\right])^2}. \tag{4}$$

It is commonly used to assess deviations from normalcy. Building upon this, Mardia [25] extended kurtosis to a *multivariate* context:

$$\mathbb{E}\left[((X-\mu)^\top \Sigma^{-1}(X-\mu))^2\right], \tag{5}$$

where $X$ denotes a multi-dimensional random vector with $\mu, \Sigma$ representing the mean and covariance matrix of $X$, respectively.

## 3 Methodology

In this section, we detail our proposed framework (InMu-Net) for MID, the architecture of which is depicted in Figure 3.

Concretely, we start by concatenating the features extracted from each modality (§2.2) to create an initial fused feature $f$, which retains comprehensive information from each modality. Then, We present the underlying motivations for our proposed supervision (§3.1). After that, we apply a denoising bottleneck module (§3.2) to perform information distillation, culminating in the refined feature $\tilde{f}$. Notably, we incorporate a saliency preservation loss (§3.3) to ensure no critical intent-relevant information is discarded. Additionally, we introduce a kurtosis regulation loss (§3.4) inspired by neural synergy [5] to represent unimodal and multimodal features more effectively against long-tail distribution.

## 3.1 Supervision Motivation

As discussed above, we aim to eliminate redundancies and noise in the visual and audio modalities within MID. To this end, we resort to information bottleneck (IB) [24], which can find the concise and compressed representation of the input. By applying the IB principle, the model can learn to filter out noisy and redundant information that might otherwise interfere with prediction accuracy. Mathematically, consider the encoded fused feature $\tilde{f}$, derived from the direct fused feature $f$. Our objective is for $\tilde{f}$ to encapsulate only

intent-relevant information while eliminating intent-irrelevant redundancies. Drawing on the principles of mutual information [9] in IB, the information between $\tilde{f}$ and $f$ can be quantified as follows:

$$\begin{aligned}
\mathcal{I}(f;\tilde{f}) = \mathcal{I}(\tilde{f};f) &= \mathbb{E}_{p(f,\tilde{f})}\left[\log\frac{p(f,\tilde{f})}{p(f)p(\tilde{f})}\right] \\
&= \iint p_{f,\tilde{f}}(f,\tilde{f})\log\frac{p_{f,\tilde{f}}(f,\tilde{f})}{p_f(f)p_{\tilde{f}}(\tilde{f})}dfd\tilde{f}.
\end{aligned} \tag{6}$$

This expression can be further expanded by introducing the target variable $y$ and applying the chain rule as follows:

$$\begin{aligned}
\mathcal{I}(f;\tilde{f}) &= \iiint p(f,\tilde{f}|y)p(y)\log\frac{p(f,\tilde{f}|y)p(y)}{p(f)p(\tilde{f})}\,df\,d\tilde{f}\,dy \\
&= \iiint p(f,\tilde{f},y)\log\frac{p(f,\tilde{f}|y)}{p(f|y)p(\tilde{f}|y)}\,df\,d\tilde{f}\,dy \\
&\quad + \iint p(\tilde{f},y)\log\frac{p(\tilde{f}|y)}{p(\tilde{f})}\,d\tilde{f}\,dy \\
&= \mathcal{I}(f;\tilde{f}|y) + \mathcal{I}(\tilde{f};y),
\end{aligned} \tag{7}$$

which distinguishes the intent-irrelevant information $\mathcal{I}(f;\tilde{f}|y)$ from the intent-relevant information $\mathcal{I}(\tilde{f};y)$. Thus, the goals of IB are to ❶ minimize the $\mathcal{I}(f;\tilde{f}|y)$ while ❷ maximize the $\mathcal{I}(\tilde{f};y)$:

$$\min_{\tilde{f}}\mathcal{L}_{IB} = \mathcal{I}(f;\tilde{f}|y) - \gamma\mathcal{I}(\tilde{f};y), \tag{8}$$

in which $\gamma$ is a scalar that determines the weight of the intent-relevant information constraint $\mathcal{I}(\tilde{f};y)$ during optimization.

However, conducting a min-max game as formulated in Eq. (8) is challenging [27, 30] due to the well-documented difficulty in estimating mutual information in high-dimensional spaces [29]. To address this issue, we introduce a denoising bottleneck to achieve goal ❶ and saliency preservation to achieve goal ❷. Additionally, we implement kurtosis regulation to ensure smooth representations across both unimodal and multimodal contexts, thereby mitigating the negative effects associated with tailed intents.

## 3.2 Denoising Bottleneck

Considering three encoded features $f_{\{t,v,a\}}$ from distinct modalities (§2.2) in MID, we first generate an initial fused feature $f = [f_t, f_v, f_a] \in \mathbb{R}^n$, where $[\cdot, \cdot]$ denotes concatenation operation. Subsequently, we suggest automatic denoising at the feature level. To elaborate, the denoising bottleneck module comprises two linear projection layers along with dropout and ReLU activation functions:

$$f \xrightarrow[\text{Dropout+ReLU}]{\text{Linear Projection}} z \xrightarrow[\text{Dropout+ReLU}]{\text{Linear Projection}} \tilde{f}, \qquad (9)$$

where $z$ is of dimension $p < n$, while the final feature representation $\tilde{f}$ retains $n$ dimensions. The re-projection of $z$ back to the same dimensions as $f$ serves two purposes: ❶ It facilitates feature-level supervision, enabling the learning of a more intent-relevant feature $\tilde{f}$ as discussed in the subsequent subsection. ❷ Empirically, we observed that the feature-level supervision proves more effective when $\tilde{f}$ and the initial feature $f$ are aligned.

**REMARK** 1. *There are several applications of the IB within the multimodal community. [11] explored the application of IB at different stages of fusion, while [52] implemented IB at the sequence level. Our method offers several advantages: ❶ simplicity in fusion approach; ❷ flexibility in handling variable-length sequence inputs. Additionally, direct comparisons are provided in the experiments (cf. Table 8).*

## 3.3 Saliency Preservation

The denoising bottleneck constrains information flow across modalities to filter out redundancy and noise. However, it might inadvertently lead to the loss of vital information as well [11]. Inspired by the success of mutual information in applications within the computer vision community [8, 22, 41], we introduce saliency preservation, which is designed to explicitly maximize intent-relevant information $\mathcal{I}(\tilde{f}; y)$ in Eq.(8), ensuring that the process of information reduction does not compromise the quality of crucial information.

**THEOREM** 1. *To elucidate, maximizing $\mathcal{I}(\tilde{f}; y)$ can be interpreted as minimizing the difference in the mutual information between the original and the denoised features for $y$, formally expressed as:*

$$\min \mathcal{I}(f; y) - \mathcal{I}(\tilde{f}; y) \iff \min H(y|f) - H(y|\tilde{f}), \qquad (10)$$

*where $H(y|f)$ is defined as the conditional entropy:*

$$H(y|f) := -\int p(f) d_f \int p(y|f) \log p(y|f) d_y. \qquad (11)$$

Building on this theoretical foundation, we further validate the effectiveness of $\tilde{f}$ through the following corollary:

**COROLLARY** 1. *If the KL-divergence [18] between the predicted distributions of the fused feature $f$ and the denoised $\tilde{f}$ equals to 0, then $\tilde{f}$ is sufficient for $y$ as well i.e.,*

$$D_{KL}\left[p(y|f)||p(y|\tilde{f})\right] = 0 \implies H(y|f) - H(y|\tilde{f}) = 0, \qquad (12)$$

*where $p(y|f), p(y|\tilde{f})$ represent the predicted distributions under $f$ and $\tilde{f}$, respectively; and $D_{KL}(\cdot)$ denotes the KL-divergence.*

To operationalize this theory, we finally frame our saliency preservation loss as follows:

$$\mathcal{L}_{saliency} = D_{KL}\left[p(y|f)||p(y|\tilde{f})\right]. \qquad (13)$$

## 3.4 Kurtosis Regulation

In multi-sensory neural processing, around 20% of the neurons account for 80% of the information propagation in cortical circuits [28]. Several researches by [7, 42] demonstrated that multi-modal computation tends to concentrate in such local cortical clusters and found significantly lower kurtosis in such clusters. Besides, low kurtosis helps alleviate sensitivity to rare events, as discussed in §2.4. These all suggest that individual unimodal and multi-modal representations in MID should exhibit low levels of kurtosis. To maintain this characteristic, we regulate the multivariate kurtosis by plugging in standard estimators for the mean and covariates:

$$\mathcal{L}_{kurtosis} = \frac{1}{N} \sum_{j=1}^{N} \left[ ((f_j - \bar{f})^\top S^{-1} (f_j - \bar{f}))^2 \right], \qquad (14)$$

where $N$ denotes the number of features, $f_j$ represents samples from features, including both unimodal features like $f_{\{t,v,a\}}$ and the direct fused feature $f$. $\bar{f}$ denotes the empirical mean feature $\bar{f} = \frac{\sum_{j=1}^{n} f_j}{N}$ and $S$ signifies the empirical covariance matrix:

$$S = \frac{\sum_{i=1}^{N} (f_i - \bar{f})(f_i - \bar{f})^\top}{N - 1}. \qquad (15)$$

Since $\tilde{f}$ is derived from $f$, we do not perform regularization on $\tilde{f}$. Note that the covariance matrix is computed via a decaying moving average over a window of multiple batches to produce smoother estimates before the inversion operation.

## 3.5 Overall Objective

Eventually, the overall loss function in our INMU-NET is:

$$\mathcal{L} = \underbrace{\mathcal{L}_f(y, y'_f) + \mathcal{L}_{modality}(y, y'_{f_t}) + \mathcal{L}_{\tilde{f}}(y, y'_{\tilde{f}})}_{\text{Foundational Supervision}} \qquad (16)$$
$$+ \alpha \mathcal{L}_{saliency}(y'_f, y'_{\tilde{f}}) + \beta \mathcal{L}_{kurtosis},$$

where $\mathcal{L}_f$ and $\mathcal{L}_{\tilde{f}}$ are respective losses to supervise the direct fused feature and denoised feature, and $\mathcal{L}_{modality}$ supervises core textual modality encoder since textual information dominates across modalities [15, 40]. $\alpha$ and $\beta$ are trade-off hyper-parameters. $y'_f$ and $y'_{\tilde{f}}$ are the classifier results from the direct fused feature and denoised feature, respectively. Note that the first three losses are directly supervised by $y$ and serve as the foundational supervision of the overall framework in the form of cross-entropy losses.

During inference, the denoised feature $\tilde{f}$ is employed to determine the ultimate intent. Consequently, INMU-NET serves as an augmentation during the training phase, with only the *denoising bottleneck module* invoked during inference, given that simple Linear and Dropout contribute minimally to inference latency.

## 4 Experiments

### 4.1 Datasets and Metrics

We conduct experiments on two benchmarks to evaluate the proposed INMU-NET: ❶ **MIntRec** [60],[1] which is a fine-grained dataset for multi-modal intent recognition. It comprises 2,224 high-quality

---

[1]https://github.com/thuiar/MIntRec/tree/main

| Model | MIntRec | | | | MELD-DA | | | |
|---|---|---|---|---|---|---|---|---|
| | ACC | wF1 | wP | R | ACC | wF1 | wP | R |
| MAG-BERT [32] | 72.65 | 72.16 | 72.53 | 69.28 | 60.63 | 59.36 | 59.80 | 50.01 |
| MulT [44] | 72.52 | 72.31 | 72.85 | 69.24 | 60.36 | 59.01 | 59.44 | 49.93 |
| MISA [12] | 72.29 | 72.38 | 73.48 | 69.24 | 59.98 | 58.52 | 59.28 | 48.75 |
| TCL-MAP [61] | 73.62 | 73.31 | 73.72 | 70.50 | 61.75 | 59.77 | 60.33 | 50.14 |
| SDIF-DA* [14] | 73.90 | 73.93 | 73.96 | 71.61 | 61.31 | 58.01 | 60.93 | 49.96 |
| InMu-Net (Ours) | **76.05**$^\dagger$ | **75.96**$^\dagger$ | **76.18**$^\dagger$ | **73.93**$^\dagger$ | **63.78**$^\dagger$ | **61.64**$^\dagger$ | **63.40**$^\dagger$ | **52.31**$^\dagger$ |

**Table 1: Experimental results on two MID datasets. Best scores are in bold and second-best scores are in underlined. Results with ∗ are obtained by re-implemented, while others are taken from the corresponding published paper. † denotes the significant paired t-tests of our InMu-Net over the baseline models at $p$-value < 0.05.**

| Model | Common | | | | | Long-tail | | | | |
|---|---|---|---|---|---|---|---|---|---|---|
| | Complain | Inform | Praise | Apologise | Thank | Agree | Flaunt | Oppose | Ask for help | Joke |
| MAG-BERT | 67.65 | 71.00 | 86.03 | 97.76 | 96.52 | 91.60 | 47.09 | 33.97 | 64.44 | 37.54 |
| MulT | 65.48 | 70.85 | 84.72 | 97.93 | 96.83 | 92.23 | 48.91 | 34.68 | 69.12 | 33.95 |
| MISA | 63.91 | 70.18 | 86.63 | 97.78 | **98.03** | 92.05 | 46.44 | 36.15 | 67.57 | 38.74 |
| TCL-MAP | **68.70** | **72.80** | 87.20 | 97.70 | 97.00 | 93.10 | 50.80 | 35.90 | 66.40 | 29.00 |
| SDIF-DA* | 67.76 | 71.24 | 87.67 | 98.11 | 97.96 | 92.31 | 44.44 | 30.02 | 67.02 | 45.56 |
| InMu-Net | 67.76 | 71.09 | **89.25** | **98.63** | **98.03** | **94.42** | **56.52** | **41.38** | **69.58** | **55.38** |
| | ↑ −0.94 | ↑ −1.71 | ↑ 1.58 | ↑ 0.52 | ↑ 0.00 | ↑ 1.32 | ↑ 5.72 | ↑ 5.23 | ↑ 0.46 | ↑ 9.82 |
| w/o $\mathcal{L}_{kurtosis}$ | 67.40 | 70.79 | 89.01 | 98.24 | 97.25 | 93.38 | 48.42 | 36.80 | 66.83 | 48.69 |
| | ↓ 0.36 | ↓ 0.30 | ↓ 0.24 | ↓ 0.39 | ↓ 0.78 | ↓ 1.04 | ↓ 8.10 | ↓ 4.58 | ↓ 2.75 | ↓ 6.69 |
| Human$^\natural$ | 80.08 | 79.69 | 93.44 | 96.15 | 96.90 | 87.21 | 78.10 | 69.04 | 88.54 | 72.22 |

**Table 2: F1-score comparison of common and long-tail subsets on MIntRec. ↑ *Number* denotes the improvement our method achieves in the current category compared to the best baseline; ↓ *Number* represents the decrease in model performance across different intent categories after removing the proposed Kurtosis Regulation. Results with ♮ are taken from Zhou et al. [61].**

samples across text, visual and audio modalities, distributed among 20 intent categories. The dataset is divided into 1,334 training samples, 445 validation samples, and 445 testing samples. ❷ **MELD-DA** [35],[2] which is a large-scale dataset designed for dialogue act classification. It includes 9,988 multi-modal samples annotated across 12 common dialogue act labels, with a split of 6,991 training samples, 999 validation samples and 1,998 testing samples.

Following previous works, we employ accuracy (ACC), weighted F1-score (wF1), weighted precision (wP), and recall (R) as evaluation metrics to assess the proposed InMu-Net framework. To account for category imbalances, the wF1 and wP metrics are calculated as weighted averages, with weights corresponding to the sample counts in each category. Unless specified otherwise, higher values indicate better performance across all metrics in this work.

### 4.2 Implementation Details

For a fair comparison, we follow Huang et al. [14], Zhang et al. [60] to adopt bert-base-uncased [16] and wav2vec2-base-960h [1] from Huggingface Library [51] to extract text and audio features and Faster R-CNN [33] from Torchvision Library to extract visual features. AdamW [23] is utilized as the optimizer with a learning rate searched from $[1e^{-6}, 3e^{-5}]$. The batch size is set as 16 for training and 8 for validation/testing. For hyper-parameter $\alpha$ and $\beta$, we test them in the range from 0.2 to 1.0 on the validation set and

choose the best-performing one to the test set, respectively. Paired t-test is performed to test the significance of performance improvement with a default significance level of 0.05. All experiments are conducted on one single NVIDIA GeForce RTX 3090. The results reported in all experiments are averages of 5 random runs.

### 4.3 Main Results

We compare InMu-Net with a series of competitive MID baselines, including: MAG-BERT [32], MulT [44], MISA [12], TCL-MAP [61] and SDIF-DA [14]. The main results on two benchmarks are reported in Table 1, from which we have the following observations:

❶ Our proposed InMu-Net consistently outperforms all baselines on both MIntRec and MELD-DA datasets and achieves a new SOTA performance. Specifically, on MIntRec dataset, it overpasses the previous SOTA model SDIF-DA by 2.91% and 2.75% on ACC and wF1, respectively; on MELD-DA dataset, it overpasses TCL-MAP by 3.29% and 3.13% on ACC and wF1, respectively. This verifies the effectiveness of InMu-Net in the MID task. Furthermore, the significance tests of InMu-Net over the baseline models show that our InMu-Net significantly outperforms the baseline models (the results of $p$-value on all evaluation metrics are less than 0.05).

❷ Notably, the gains on MELD-DA are more pronounced. We suppose the reason is that MELD-DA is more challenging, whose data redundancy problem is more serious, involving complex scenes

---
[2]https://github.com/thuiar/TCL-MAP

| Setting | $\mathcal{L}_f$ | DB | $\mathcal{L}_{\tilde{f}}$ | $\mathcal{L}_{saliency}$ | $\mathcal{L}_{modality}$ | $\mathcal{L}_{kurtosis}$ | MIntRec | | | | MELD-DA | | | |
|---|---|---|---|---|---|---|---|---|---|---|---|---|---|---|
| | | | | | | | ACC | wF1 | wP | R | ACC | wF1 | wP | R |
| (a) | ✓ | - | - | - | - | - | 69.68 | 68.40 | 68.84 | 67.81 | 58.25 | 55.03 | 57.26 | 47.98 |
| (b) | ✓ | - | - | - | $t$ | ✓ | 71.49 | 70.31 | 70.75 | 69.99 | 60.51 | 58.28 | 60.42 | 50.16 |
| (c) | ✓ | ✓ | ✓ | - | $t$ | ✓ | 74.12 | 72.87 | 73.22 | 72.18 | 62.59 | 59.50 | 61.67 | 51.35 |
| (d) | ✓ | ✓ | ✓ | ✓ | $t$ | - | 74.40 | 73.13 | 73.50 | 72.46 | 62.94 | 59.69 | 61.91 | 51.67 |
| (e) | ✓ | ✓ | ✓ | ✓ | - | ✓ | 72.81 | 71.48 | 71.89 | 70.88 | 60.98 | 57.92 | 60.04 | 50.34 |
| (f) | ✓ | ✓ | ✓ | ✓ | $a$ | ✓ | 73.18 | 71.82 | 72.26 | 71.25 | 61.27 | 58.17 | 60.44 | 50.62 |
| (g) | ✓ | ✓ | ✓ | ✓ | $v$ | ✓ | 73.07 | 71.75 | 72.21 | 71.10 | 61.12 | 58.11 | 60.38 | 50.53 |
| (h) | ✓ | ✓ | ✓ | ✓ | $t$ | ✓ | **76.05** | **75.96** | **76.18** | **73.93** | **63.78** | **61.64** | **63.40** | **52.31** |

**Table 3: Ablation studies. "DB" is short for denoising bottleneck. "$t, a, v$" denotes textual, audio, and visual modality, respectively.**

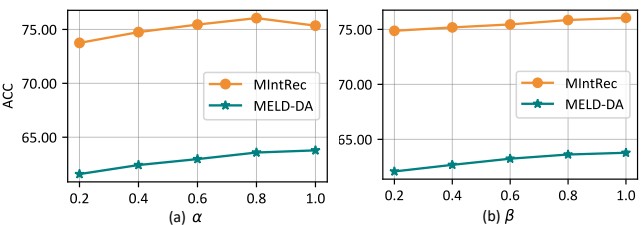

**Figure 4: Hyper-parameter analyses. Effect of the trade-off hyper-parameter (a) $\alpha$ and (b) $\beta$ in Eq.(16).**

| Model | Perplexity ($\downarrow$) |
|---|---|
| TCL-MAP [61] | 2.55 |
| SDIF-DA [14] | 2.48 |
| INMU-NET | 1.82 |

**Table 4: Comparison with SOTA baselines on perplexity. $\downarrow$ denotes lower is better.**

and overlapping characters. The proposed INMU-NET excels in denoising and preserving vital information, coupled with enhanced robustness via kurtosis regulation, resulting in superior performance over strong baseline models TCL-MAP and SDIF-DA.

❸ To clearly demonstrate how our method addresses the long-tail distribution issue, we selected the top five and bottom five intents based on their occurrence frequencies in the training set. These are referred to as the "Common" and "Long-tail" subsets, respectively, and their F1 scores are presented in Table 2. We find our INMU-NET achieves superior performance in 8 out of the 10 intent categories, particularly showing notable improvements in the "Long-tail" intents. Thanks to the kurtosis regularization, INMU-NET can adaptively mitigate the adverse effects of the long-tail distribution, achieving robust performance.

### 4.4 Ablation Study

We perform a set of ablation studies to understand the necessity of the different designs and strategies in the proposed INMU-NET. From the results in Table 3, we can obtain the following takeways:
**Denoising Bottlneck.** The denoising bottleneck module coupled with saliency preservation loss is first removed to not perform information filtering. By comparing the row setting (b) to setting (h) in Table 3, we can conclude that the decreased performance implies that the proposed denoising bottleneck actually contributes to eliminating redundancy and noise within the modalities.
**Saliency Preservation.** There is a consistent performance degradation on both datasets (comparing setting (c) to setting (h)) when the denoising bottleneck lacks saliency preservation supervision. A plausible deduction is that the supervision mitigates the loss of valuable information due to intent-irrelevant feature filtering.
**Kurtosis Regualion.** ❶ Here, we remove the kurtosis regulation to verify its effectiveness (comparing setting (d) to setting (h)). The

poor results show that appropriate regulating multi-modal features can boost model performance. ❷ Furthermore, when comparing the data in Table 2 without the $\mathcal{L}_{kurtosis}$ term (*w/o $\mathcal{L}_{kurtosis}$*), a noticeable decline in model performance is observed. This further demonstrates that the proposed kurtosis regularization can effectively alleviate the sensitivity to tail intents and mitigate long-tail distribution issues, thereby enabling more robust predictions.
**Centre Modality.** As mentioned in §3.5, results by text-centric supervision tend to perform better as low information intensity and high redundancy in other modalities. Thus, we evaluate results based on no, audio and visual modality supervision (setting (e), (f) and (g)). We observe an obvious decline in performance when audio, visual, or no modality is used as the central supervision.

### 4.5 Hyper-parameter Analysis

Since Eq.(16) encompasses multiple loss components, we delved into the influence of the core elements as shown in Figure 4. Specifically, we select the best-performing model based on the validation set and then evaluate it on the test set: ❶ The parameter $\alpha$ indicates the importance of $\mathcal{L}_{saliency}$. We evaluate the scale range setting $\alpha \in [0.2, 1.0]$ as shown in Figure 4(a). We find that accuracy is improved and saturated with 0.8 and 1.0 on MIntRec and MELD-DA, respectively. Thus, we set $\alpha = 0.8$ for MIntRec and 1.0 for MELD-DA in practice. ❷ The parameter $\beta$ signifies the extent of involvement of $\mathcal{L}_{kurtosis}$. Our observations indicate a relative insensitivity to parameter selection, as the incorporation of kurtosis regulation generally yields gains for MID, albeit to varying degrees. As a result, we adopt $\beta = 1.0$ to achieve the best performance.

### 4.6 Perplexity Evaluation

Since the denoising module plays an important role in the proposed approach, we provide more insight analysis about it. Specifically, we evaluated the perplexity for the prediction of the golden label across two state-of-the-art baselines (*i.e.*, TCL-MAP and SDIF-DA) and our proposed InMu-Net on the MIntRec test set. From the results in

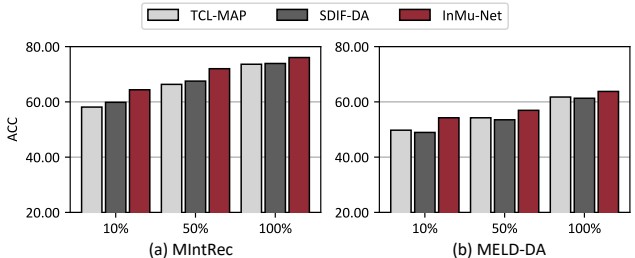

**Figure 5: Performances on low-resource settings.**

| Modality | | | MIntRec | | MELD-DA | |
|---|---|---|---|---|---|---|
| t | a | v | ACC | wF1 | ACC | wF1 |
| - | ✓ | ✓ | 58.48 | 57.27 | 47.43 | 44.96 |
| ✓ | ✓ | - | 68.76 | 67.35 | 57.02 | 54.13 |
| ✓ | - | ✓ | 70.14 | 68.98 | 58.54 | 55.47 |
| ✓ | ✓ | ✓ | **76.05** | **75.96** | **63.78** | **61.64** |

**Table 5: Results on modality corruption.**

Table 4, we observe that both TCL-MAP and SDIF-DA exhibit perplexity around 2.5, whereas our method demonstrates a significant reduction (-0.7). This compellingly illustrates the effectiveness of our denoising bottleneck module in accurately predicting intents.

## 4.7 Low-resource Settings

To investigate the effect and robustness of INMU-NET in low-resource scenarios, we conduct experiments using different limited training sizes following Huang et al. [14]. From the results in Figure 5, we observe that INMU-NET consistently outperforms SOTA baselines, especially when the resource is quite limited (10%). This indicates that INMU-NET can sustain denoising capabilities within resource-constrained scenarios and distill intent-relevant information from multi-modal representations, thus boosting performance.

## 4.8 Modality Corruption

We further assess the model performance by removing one modality at a time. From the results in Table 5, we find that ❶ the tri-modal combination yields the highest performance, indicating that our INMU-NET can learn complementary information from different modalities. ❷ the performance drops sharply when the textual modality is removed. We attribute this to the fact that textual modality has higher information density compared to redundant audio and visual modalities. This underscores two critical insights: First, eliminating noise and redundancy to enhance the information density of visual and audio modalities is crucial during fusion. Second, text-central fusion results may help boost performance in MID.

## 4.9 Generalizability Analysis

**Analysis of Cross-architecture Scenario**. To evaluate the generalizability of our proposed INMU-NET, we conduct preliminary experiments on two representative baselines MuIT and SDIF-DA. To be specific, we retain the baseline loss function for a fair comparison, integrating only our three vital components and corresponding supervisions on the fused feature. The results depicted in Table 6

| Model | MIntRec | | MELD-DA | |
|---|---|---|---|---|
| | ACC | wF1 | ACC | wF1 |
| MulT* [44] | 72.23 | 71.98 | 60.27 | 58.95 |
| with INMU-NET | **74.01** | **73.85** | **61.53** | **59.75** |
| SDIF-DA* [14] | 73.90 | 73.93 | 61.31 | 58.01 |
| with INMU-NET | **75.25** | **75.08** | **62.94** | **60.38** |

**Table 6: Results on cross-architecture scenarios.**

| Model | Latency/Inference Time per Sample | Speedup |
|---|---|---|
| TCL-MAP [61] | 25.8ms | 1.0x |
| SDIF-DA [14] | 25.4ms | 1.0x |
| INMU-NET | 25.5ms | 1.0x |

**Table 7: Comparison on computation (inference) latency.**

indicate that baselines augmented with INMU-NET outperform their original counterparts. This verifies our work's contribution is orthogonal to theirs, considering changing the architecture of INMU-NET for better multi-modal fusion is still a promising avenue.

**Analysis of Cross-task Scenario**. We further conduct comparison experiments on multi-modal sentiment analysis (MSA) to evaluate the proposed INMU-NET and the results are reported in Table 8. It can be observed that INMU-NET achieves competitive performance compared with MSA baselines, which verifies that INMU-NET can generalize the denoising ability to different tasks. Note that MMIM performs hierarchical mutual information maximization for each modality, which is deeply integrated with the overall framework, and proposes a parameterized method to approximate the true value. Whereas, even without careful hyper-parameters tuning, our proposed INMU-NET outperforms MMIM on 4 out of 6 metrics and achieves similar performance on the remaining two metrics.

## 4.10 Computation Efficiency

As the proposed method is applicable to real-world applications, we conducted a preliminary latency comparison among the two most advanced baseline models and our method, with the results reported in Table 7. We find that our model's inference speed surpasses that of TCL-MAP and maintains a speedup on par with current SOTA methods. This can be attributed to the fact that our method functions as an augmentation during the training phase, with only the denoising bottleneck module being activated during inference. Given that simple Linear and Dropout layers contribute minimally to the inference latency, this ensures efficient performance.

## 4.11 Visualization

To qualitatively demonstrate how our proposed INMU-NET filters noise and redundancy while capturing precise intent-relevant information, we provide GradCAM-CAM [36] visualizations of INMU-NET and SOTA baseline SDIF-DA. From Figure 6 case (a), it can be seen that while both models SDIF-DA and INMU-NET successfully identify the critical term "*gift*" in the visual modality. However, the capture range of SDIF-DA is broader and more diffuse, including irrelevant background details. In contrast, our proposed INMU-NET focuses narrowly and precisely on the gift on the table, demonstrating its ability to filter out extraneous information and pinpoint

| Model | MOSI [57] | | | MOSEI [58] | | |
|---|---|---|---|---|---|---|
| | MAE($\downarrow$) | Corr($\uparrow$) | Acc-7($\uparrow$) | MAE($\downarrow$) | Corr($\uparrow$) | Acc-7($\uparrow$) |
| ICCN [38] | 0.860 | 0.710 | 39.0 | 0.565 | 0.713 | 51.6 |
| MISA [12] | 0.783 | 0.761 | 42.3 | 0.555 | 0.756 | 52.2 |
| Self-MM [56] | 0.712 | 0.795 | 45.8 | 0.529 | 0.767 | 53.5 |
| MMIM [11] | 0.700 | 0.800 | 46.7 | 0.526 | 0.772 | 54.2 |
| DBF [52] | **0.693** | **0.801** | 44.8 | 0.523 | 0.772 | 54.2 |
| INMU-NET (Ours) | 0.694 | 0.798 | **46.9** | **0.520** | **0.774** | **54.5** |

Table 8: Results of the experiments on cross-task scenarios. $\downarrow$ denotes lower is better. $\uparrow$ denotes higher is better.

**(a)** *so thank you all so much for my gifts*        **(b)** *this place is crawling with raccoons*

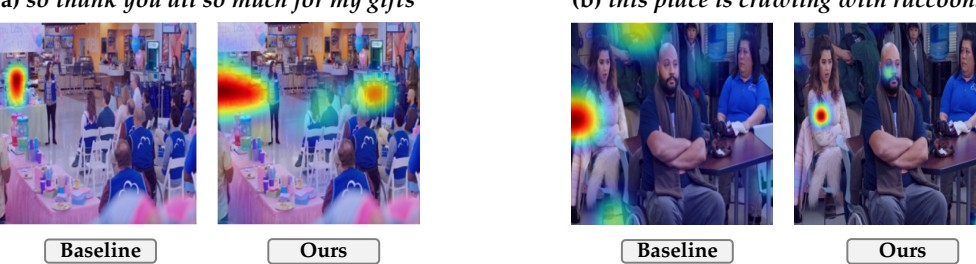

Figure 6: Comparison of Grad-CAM [36] visualizations between SOTA baseline SDIF-DA and our INMU-NET.

intent-relevant details with high accuracy. A similar situation can be observed in Figure 6 case (b), where the baseline model SDIF-DA exhibits excessive redundant attention, while our INMU-NET accurately captures the woman speaking in the figure.

## 5 Related Work

**Multi-modal Intent Detection.** Multi-modal intent detection (MID) is a significant task for understanding human language in task-oriented dialogue systems [2–4, 62–64]. Compared to text-only intent detection, which ascertains the objectives of users conveyed through their utterances, MID integrates facial expressions and audio signals to fully leverage the complementary and interactive information provided by diverse modalities. A series of models [14, 35, 60, 61] have been proposed and made promising progress. Therein, Zhou et al. [61] introduced a token-level contrastive learning coupled with modality-aware prompting to improve modality fusion. Concurrently, Huang et al. [14] developed a Transformer-based [46, 47] framework that progresses from shallow to deep interactions, complemented by ChatGPT-based data augmentation techniques to align features across modalities.

Our work do not focus on intricate fusion mechanisms [39] or architectural intricacies, our approach emphasizes the perspective of multi-modal information and data distribution.

**Information Bottleneck.** The InfoMax proposed by Linsker [20] seeks to maximize the mutual information between feature and model output. Along this way, there have been many works that explore optimal ways for mutual information estimation. Han et al. [11] built up a hierarchical mutual information maximization guided model for multi-modal sentiment analysis. Wu et al. [52] focused on video-based sentiment analysis and performed contrastive learning to achieve mutual information maximization. Despite these advancements, these theories in MID remain under-studied.

In contrast to existing works, our INMU-NET devises a denoising bottleneck and a saliency preservation loss to precisely filter intent-irrelevant information and keep intent-relevant information in an adaptive manner, which obviates the reliance on heuristic or greedy feature selection methods [19, 52] and coarse-grained gate filtering mechanism [53, 55]. Furthermore, the proposed feature-level denoising offers advantages in handling variable-length inputs.

**Multi-sensory Processing.** In multi-sensory processing research area, different sensory modalities are processed individually and then combined in various multimodal convergence zones, including cortical and subcortical regions [10]. Several research such as Faber et al. [7], Timme et al. [42] demonstrated that multi-modal computation tends to concentrate in such local cortical clusters and found significantly lower kurtosis in such clusters.

In this work, we resort to multivariate kurtosis [34] to alleviate the long-tail distribution problem in MID. The most straightforward remedy to this problem is to rebalance the training dataset through weighted sampling. However, it is a suboptimal strategy that may be detrimental to the accuracy of the head classes [49]. Innovatively, we treat the fused multi-modal feature as cortical clusters, controlling peaking to reduce sensitivity to tailed intents.

## 6 Conclusion

In this paper, we proposed INMU-NET, a new framework from the information bottleneck and multi-sensory processing perspectives, to tackle modality redundancy and long-tailed distribution of labels in MID jointly. INMU-NET maximizes the intent-relevant information in fused multi-modal features by the proposed *denosing bottleneck* and minimizes the intent-irrelevant information by the proposed *saliency preservation* loss. Moreover, *kurtosis regulation* is introduced to reduce the negative impact of long-tail distributions. Extensive experiments and analyses on two MID benchmarks show the superiority of our proposed framework.

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
