# OpenReview forum: "InMu-Net: Advancing Multi-modal Intent Detection via Information Bottleneck and Multi-sensory Processing"
_acmmm.org/ACMMM/2024/Conference — MM2024 Oral_

### Official Review · Reviewer_eAJz · 2024-05-20

**Rating:** 3
**Confidence:** 1

**Summary:**

This paper propose InMu-Net, a novel framework for Multi-modal intent detection that tackles modality redundancy and long-tailed distribution of labels in MID using information bottleneck and multi-sensory processing perspectives.

**Strengths:**

This paper proposes a novel framework for Multi-modal intent detection that introduces a denoising bottleneck module, saliency preservation loss and kurtosis regulation method.

The authors conduct comprehensive experiments on MID benchmark datasets, and the results reveal the potential and robustness of InMu-Net.

**Limitations:**

•	In Table 2, the results of “Flaunt”, ”Opposes” and “Joke” in the “Long-tail” subsets have huge improvement, but the improvement of “Agree” and “Ask for help” is  relatively low and equivalent to the improvement of “Praise” and “Apologies” in “Common” subsets. This makes it less convincing that InMu-Net mitigate the adverse effects of the long-tail distribution since it may come from the improvement of the model's detection ability for certain scenarios rather than being more adaptable to long tailed distributions.

•	Only one visualization result is provided, it is recommended to present more visualization results for comparison.

•	Appendix for more details of analysis of cross-task scenario(L799) is not provided.

**Suitability:**

2

---

### Official Review · Reviewer_xfxE · 2024-05-24

**Rating:** 5
**Confidence:** 2

**Summary:**

The paper introduces InMu-Net, a framework for multi-modal intent detection (MID). The authors propose a denoising bottleneck module to filter out irrelevant information, a saliency preservation loss to ensure relevant information is retained, and kurtosis regulation mitigate issues from long-tail distributions. The results show that InMu-Net outperforms state-of-the-art models on two MID benchmark datasets.

**Strengths:**

1.	The paper provides an extensive evaluation of the model across multiple benchmarks and conditions, including low-resource scenarios and modality corruption tests.

2.	The introduction of kurtosis regulation to handle long-tail distributions is novel. This component helps the model perform better on underrepresented long-tail subset.

**Limitations:**

1.	The design of denoising bottleneck is not novel, which can be seen as a simple denoising autoencoder on the feature level.

2.	There are two hyperparameters $\alpha$ and $\beta$ for the saliency loss and Kurtosis loss. However, the weights for other loss components are set to 1. The reason for these setups is not clearly discussed.

**Suitability:**

3

---

### Official Review · Reviewer_Vp2m · 2024-05-26

**Rating:** 5
**Confidence:** 3

**Summary:**

This paper aims to address the multi-modal intent detection task. It proposes InMu-Net to tackle the challenges that noise and redundancy in different modalities and long-tailed distributions of intent categories. A denoising bottleneck is constructed to distill information for mutual information estimation and maximize the intent-relevant information between the modal-fused feature and target intent label. Besides, a kurtosis regulation loss is designed to alleviate the negativity of tailed intents. Extensive experiments demonstrate the effectiveness of the InMul-Net in different benchmarks and its ability of denoising and robustness.

**Strengths:**

1. The motivation of this paper is well-founded and clear, and the theoretical support of the model is sufficient and convincing.
2. The experiments are sufficient and have demonstrated the effectiveness of the proposed method.

**Limitations:**

1. As  is a 1D vector and  is a intent label, how to calculate their joint probability distribution is not mentioned.
2. The detailed formula of loss  and  is not mentioned.

**Suitability:**

3

---

### Meta-Review · Area_Chair_EMRF · 2024-07-11

**Recommendation:** Accept (Oral)
**Confidence:** 4

**Metareview:**

**Conclusion: Accept as an Oral Paper**

The paper presents InMu-Net, a framework designed to address challenges in multi-modal intent detection (MID), such as noise, redundancy in different modalities, and long-tailed distributions of intent categories. The reviewers have provided positive feedback on several fronts, including the well-founded motivation, extensive experimentation, and innovative approach. Based on these merits, I recommend this submission for acceptance as an Oral Paper.

**Strengths:**

1. **Motivation and Theoretical Support:** The paper is well-motivated with clear theoretical support, making a compelling case for its proposed methods. The framework's components, such as the denoising bottleneck and kurtosis regulation, are theoretically sound and convincingly argued.

2. **Comprehensive Experiments:** Extensive evaluations across multiple benchmarks demonstrate the effectiveness and robustness of InMu-Net. The model performs well in various conditions, including low-resource scenarios and modality corruption tests, highlighting its practical applicability.

3. **Novelty in Handling Long-Tailed Distributions:** The introduction of kurtosis regulation to address the issue of long-tailed distributions is innovative. This component significantly enhances the model's performance on underrepresented categories, which is a notable contribution to the field.

**Weaknesses:**

1. **Details of Joint Probability Distribution Calculation:** The paper does not provide sufficient details on how the joint probability distribution between the 1D vector and intent label is calculated. This omission leaves a gap in understanding the full implementation of the proposed method.

2. **Lack of Novelty in Denoising Bottleneck Design:** The design of the denoising bottleneck is seen as a straightforward application of a denoising autoencoder at the feature level. This could be perceived as lacking novelty in comparison to existing methods.

3. **Limited Visualization and Appendix Details:** Only one visualization result is presented, and the appendix lacks detailed analysis of the cross-task scenario. More visual comparisons and additional information in the appendix would strengthen the paper's empirical evidence and overall presentation.

In summary, despite some areas needing further clarification and enhancement, the paper's strengths in motivation, comprehensive experimentation, and novel approach to handling long-tailed distributions make it a valuable contribution to the field of multi-modal intent detection. Therefore, it merits acceptance as an Oral Paper for ACM MM 2024.